

**Aerosol Optical Depth retrievals in Central Amazonia from a Multi-**
**Filter Rotating Shadow-band Radiometer on-site calibrated**
**Nilton E. Rosário[1], Tamara Sauini[1], Theotonio Pauliquevis[1], Henrique M. J. Barbosa[2],**
**Marcia A. Yamasoe[3], Boris Barja[4]**
[1]Universidade Federal de São Paulo - Rua São Nicolau 210 - Diadema - SP - CEP 09913-030 – Brazil
[2]Instituto de Física da Universidade de São Paulo - Rua do Matão 1371 - São Paulo - SP - CEP 05508-090 - Brazil
[3]Instituto de Astronomia, Geofísica e Ciências Atmosféricas - Universidade de São Paulo - Rua do Matão 1226 - São Paulo -
SP - CEP 05508-090 – Brazil
[4]Universidade de Magallanes - Manuel Bulnes 01855, Punta Arenas, Region de Magallanes y de la Antártica Chilena, Chile
Correspondence to: Nilton E. Rosario (nrosario@unifesp.br)
**Abstract**
Extraterrestrial spectral response calibration of a Multi-Filter Rotating Shadow band Radiometer (MFRSR) under
Amazonian Forest atmosphere pristine conditions using the Langley plot method was performed and evaluated. The
MFRSR is installed in central Amazonia as part of a long-term monitoring site, which was used in the context of the
GoAmazon2014/5 Experiment. It has been operating continuously since 2011 without regular extraterrestrial
calibration, preventing its application to accurate monitoring of aerosol particles. Once calibrated, the MFRSR
measurements were applied to retrieve aerosols particles columnar optical properties, specifically Aerosol Optical
Depth ($AOD_\lambda$) and Ångström Exponent (AE), which were evaluated against retrievals from a collocated CIMEL
sunphotometer belonging to the AErosol RObotic NETwork (AERONET). Results obtained revealed that Amazonian
pristine conditions are able to provide MFRSR extraterrestrial spectral response with relative uncertainty lower than
1.0% at visible channels. The worst estimate (air mass = 1) for absolute uncertainty in $AOD_\lambda$ retrieval varied from
~0.02 to ~0.03, depending on the assumption regarding uncertainty for MFRSR direct-normal irradiance measured
at the surface. Obtained Root Mean Square Errors (RMSE ~ 0.025) from the evaluation of MFRSR retrievals against
AERONET $AOD_\lambda$ were, in general, lower than estimate MFRSR $AOD_\lambda$ uncertainties, and close to AERONET field
sunphotometers (~ 0.02).



**1. Introduction**
Aerosol Optical Depth (AOD) is an important variable to characterize atmospheric particles
columnar abundance and is also fundamental to estimate their direct radiative forcing in the climate system
(Shaw, 1983, Kaufman et al. 2002, Menon, 2004, Satheesh and. Srinivasan, 2005). Its relevance is also
growing in the context of air quality monitoring from satellite (Hoff and Christopher, 2009, van Donkelaar
et al., 2010, van Donkelaar et al. 2013). However, the so called Extraterrestrial Response Calibration (ERC)
of the radiometers designed to monitor AOD, for instance sun tracking and shadow-band radiometers
(Holben et al., 1998, Harrison and Michalsky, 1994), is a critical issue to the accuracy of AOD retrievals
(O'Neill et al., 2005, Sinyuk et al., 2012, di Sarra et al., 2015). Therefore, regular and adequate calibration
of sun-tracking and shadow-band radiometers dedicated to monitor AOD is vital (Holben et al., 1998, Eck
et al., 1999, Michalsky et al., 2001). The ERC consists in the estimation of the solar energy that would be
measured by the instrument at the top of the atmosphere (TOA) or in hypothetical absence of the
atmosphere. It remains one of the most critical calibrations to the accuracy of AOD retrieval (Forgan, 1994;
Michalsky et al.; 2001, Eck et al., 1999; Chen et al., 2013). The classical way to perform ERC is based on
the Langley plot method, for which is recommended to take measurements on high mountains tops under
clean air and stable conditions (Shaw et al., 1976, Holben et al., 1998). However, very often, regular trip to
very high and clean mountain top to perform ERC are not possible, either due to the lack of resources or to
avoid data collection interruption. Consequently, with the spread of ground based AOD monitoring
networks, on site calibration based on multiple Langley plots has been successfully adopted elsewhere
(Michalsky et al., 2001, Augustine et al., 2008, Rosario et al., 2008, Mazzola et al., 2010, Michalsky et al.,

21 2013).

During the last decades, Amazonia has been a stage for various intensive and mid to long term
atmospheric experiments (Avissar et al., 2002, Silva Dias et al., 2002, Andreae et al., 2004, Martin et al.,
2016), performing a large number of field measurements, and regularly including ground-based monitoring
of AOD. Given the inherent complex logistics that characterize field experiments in Amazonia, regular trip
to distant clean mountain top to perform ERC of AOD monitoring devices operating inside the forest it is



a challenge, mainly for long-term sites. Unlike AErosol RObotic NETwork (AERONET) sunphotometers,
which have a regular calibration logistic supported by NASA (Holben et al., 1998), other ground-based
devices for AOD monitoring operating inside the Amazonia have to find alternative ways to provide regular
calibration. Multi-Filter Rotating Shadow-band Radiometers (MFRSR, Harrison and Michalsky, 1994) has
been also deployed recurrently in the Amazon basin to monitor spectral and broadband solar irradiance and
AOD during specifics seasons (Yamasoe and Rosario, 2009, Rosario et al., 2009, Yamasoe et al., 2014,
Martin et al., 2016), and more recently focusing in mid and long-term monitoring (Barbosa et al., 2014).
An experimental site, located in central Amazonia, and included in the context of the Observations and
Modelling of the Green Ocean Amazon (GoAmazon2014/5, Martin et al., 2016) under the reference of T0e
is operating since the year of 2011 a MFRSR as part of a set of instruments to perform long term
atmospheric monitoring of convection, radiation, aerosols and cloud properties in central Amazonia
(Barbosa et al. 2014). GoAmazon experimental sites range from time point zero (T0) upwind of pollution
associated with Manaus city, Brazil (Figure 1) to sites in the midst (T1) and downwind (T2, T3) of the
pollution plume (Martin et al., 2016). The MFRSR is being operated in central Amazonia since 2011
without performing its ERC, which prevent its application to retrieve AOD. In this context, the question
that drives the focus of the present study is: Does Amazonia pristine atmosphere conditions provide
successful scenarios for Extraterrestrial Response Calibration? Amazonia atmosphere under pristine
conditions have been denominated as Green Ocean due to its very low pollution concentration, comparable
to remote ocean areas (Robert et al., 2001, Andreae et al., 2004), which is a fundamental requirement to
apply the Langley plot method. To answer the question posed, the present paper describes and discusses
methods and results of an effort to calibrate, on site, the cited MFRSR. Its subsequent application to
characterize the AOD variability is evaluated against AOD retrievals from a collocated Cimel
sunphotometer from AERONET (Holben et al., 1998). The manuscript is organized as follow: **section 2**
describes the experimental site, a brief overview on MFRSR and Langley plot method and AOD retrieval
theory, **section 3** consists of results and discussion and final remarks are exposed in **section 4**.



**2. Experimental site, instruments, and methods**
**2.1    Experimental site T0e**
The T0e site has been operating continuously since February 2011 in Central Amazonia, up-wind from
Manaus city (59º 58' 12''W and 02º 53' 27''S, **Figure 1**), with a set of collocated atmospheric monitoring
instruments that include a MFRSR, a Cimel sunphotometer and a Raman lidar (Barbosa et al., 2014). The
site main goal is to provide long term characterization of diurnal and seasonal cycles of clouds and
convection and the interactions and feedback mechanisms between water vapour, clouds, radiation and
aerosol particles. It was incorporated as part of the GoAmazon 2014/15 experiment (Martin et al., 2016)
network sites, an international experiment designed to investigate the interactions that involve Amazonia
natural atmosphere conditions and the air pollution plume from Manaus city.
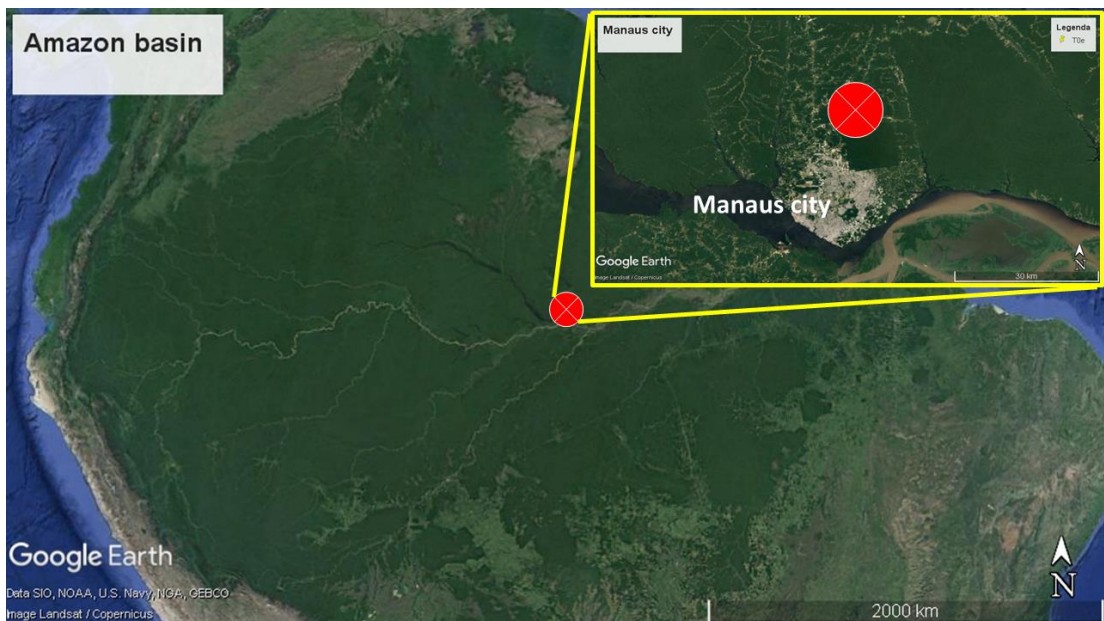

**Figure 1**- T0e site location in the Amazon basin and a zoom in showing the site location upwind of the Manaus City (source: Google Earth maps).

The GoAmazon2014/5 sites were classified from time point zero (T0) upwind of the plume, to T1 in the
midst of the plume, to T2 just downwind of the Manaus, to T3 furthest downwind of Manaus (70 km).



During the wet season, the atmosphere at T0e site is a clean reference, since its location upwind of Manaus
prevents the site of being affected by the city pollution plume. Meanwhile, during the dry season the
atmospheric column at T0e, likewise large portion of atmosphere across central Amazonia, is influenced
by smoke from biomass burning emissions that occur throughout the Amazon basin.
**2.2   Instruments**
Multifilter Rotating Shadow-band Radiometer is designed to monitor global-horizontal, diffuse-
horizontal and direct-normal solar irradiances at narrow and broadband channels (Harrison et al., 1994). It
has been used worldwide to derive columnar aerosol optical properties (Harrison and Michalsky, 1994;
Alexandrov et al., 2002; Rosario et al., 2008, Michalsky et al., 2010, Mazzola et al., 2010, Michalsky and
LeBaron, 2013), water vapour (Michalsky et al, 1995, Alexandrov et al., 2009, Schneider et al, 2010) and
cloud optical properties (Min and Harrison, 1996, Wang and Min, 2008, Kassianov et al., 2011). Direct-
normal spectral irradiance ($I_{DN,\lambda}$) at the surface, needed to perform AOD retrievals, is obtained via the
difference between global-horizontal and diffuse-horizontal irradiances divided by the cosine of the solar
zenith angle (Harrison et al., 1994). Once MFRSR angular and spectral responses are properly characterized
and the automated shadow-band system adequately adjusted, accuracy in $I_{DN,\lambda}$ is expected to be comparable
to sunphotometers (Harrison et al., 1994). However, once in field, MFRSR filters transmission may suffer
degradation with time (Mychalsky et al, 2001, Michalsky and LeBaron, 2013), which makes regular ERC
critically necessary to keep the accuracy of AOD retrievals. The MFRSR of the present study has been
operating with sporadic interruptions at T0e providing irradiances measurements at time interval of 1
minute at five narrow-band channels (415, 500, 610, 670 and 870 nm) with half-bandwidth of 10 nm and
able to permit AOD retrieval. Given the high cloud cover in central Amazonia, the MFRSR high frequency
measurements are crucial to improve the frequency of AOD retrieval under cloudy sky and, therefore,
minimizes the AERONET known bias toward clear-sky condition (Levy et al., 2010).



**2. 3 Langley plot calibration and uncertainties**
Langley plot calibration method is based on Lambert-Beer law (Shaw, 1983), which describes the
attenuation of a monochromatic beam propagating through a medium.
$$I_{DN,\lambda} = f(d)\, I_{0,\lambda}\, e^{-m\tau_\lambda} \qquad \text{eq. 1}$$
where, considering the full atmospheric column as a medium, $I_{DN,\lambda}$ is the direct solar spectral irradiance at
wavelength $\lambda$ measured at the surface by the MFRSR, $I_{o,\lambda}$ is the solar spectral irradiance that would be
measured in the absence of the atmosphere at Earth-Sun mean distance ($d_o$), $f(d)$ is a correction factor
related to Earth-Sun distance variation (Iqbal, 1983), and $m$ and $\tau_\lambda$ represent the atmosphere relative
optical air mass and total optical depth, respectively. Linearization of the equation 1 by applying the natural
logarithms to the both sides of the equation leads to a linear relation between $m$ and $\ln(I_{DN,\lambda})$, on which
$\tau_\lambda$ and $\ln(f(d)I_{o,\lambda})$ represent, respectively, the angular and linear coefficients.
$$\ln(I_{DN,\lambda}) = \ln(f(d)I_{0,\lambda}) - m\tau_\lambda \qquad \text{eq. 2}$$
Knowing $\ln(I_{DN,\lambda})$ over a range of $m$, during which atmosphere remained clean and stable, the least-
squares regression method can be applied to provide a linear fit formulation between both variables, where
the angular coefficient is the mean atmosphere optical depth, and the linear coefficient represents the case
of $m$ equal to zero, a hypothetical absence of atmosphere, from which an estimation of the solar
extraterrestrial spectral irradiance $(I_{o,\lambda})$ can be made.
In the present study, the atmosphere relative optical air mass ($m$) was calculated as a function of
Solar Zenith Angle (SZA) based on the Kasten and Young (1989) and $\ln(I_{DN,\lambda})$ taken from MFRSR direct-
normal irradiance measurements for the years of 2012 and 2015. As we assumed that both, the response
variable, $\ln(I_{DN,\lambda})$, and the predictor variable, $m$, are subject to errors, it was applied the least square
regression treatment that consider errors in both adjusted variables (Irvin and Quickenden, 1983). The errors
in $\ln(I_{DN,\lambda})$ were obtained through error propagation theory considering Harrison et al. (1994) estimate of
uncertainty to MFRSR direct-normal irradiance ($\sigma_{I_{DN,\lambda}} = 2\%$). Regarding error in the airmass ($\sigma_m$) we



based on the study of Tomasi and Petkov (2014), which compared atmospheric airmass results from Kasten
and Young (1989) formulation against rigorous calculation and found differences lower than 0.8%.
Therefore, we assume 0.8% as an estimate of uncertainty to the airmass calculated using Kasten and Young
(1989). Following previous studies suggestion (Mazzola et al., 2010 and Alexandrov et al.,2004), to apply
least square regression we adopted the airmass range from 2.0 to 5.0. For airmass larger than 5.0, high solar
energy incident angles, calibration may be affected due the uncertainty of the MFRSR cosine angle
correction and the shadow-band correction, meanwhile low airmasses, near 1.0, increase the probability of
turbulent atmospheric conditions and, therefore, the reduction of the optical depth stability (Chen et al.

9    2013).

The quality of the linear fit derived using leas-square regression is highly dependent on optical depth
temporal stability, which is more likely to be observed under aerosol background conditions and stable
atmosphere. To obtain a set of linear fit able to provide high quality Langley plot calibration samples, for
both years 2012 and 2015, were selected only morning cases, to avoid the afternoon vigorous convection,
and only linear fit with correlation coefficients ($R^2$) higher than 0.990. This is the minimal value usually
obtained for calibration performed at high mountain top (Schmid and Wehrli, 1995). Also, considering
Schafer et al. (1998) study on AOD climatology across the Amazon basin, only AOD conditions typical of
background conditions were selected. For the both years studied, 2012 and 2015, the MFRSR final
extraterrestrial spectral response calibration ($< I_{o,\lambda} >$) was estimated from the mean of the correspondent
set of extraterrestrial response calibration ($I_{o,\lambda}$) obtained from individual Langley plot calibrations. The
uncertainties of the derived final calibrations were estimate as the standard error of the mean($\sigma_{<I_{o,\lambda}>}$).
Subsequently, the final calibrations results were applied to retrieve $AOD_\lambda$ over the Toe site using the
MFRSR.



**2.4 Aerosol Optical Depth ($AOD_\lambda$) inversion and uncertainty estimate**
From the eq. 2, the atmospheric total optical depth ($\tau_\lambda$) can be separated as follow:

$$\tau_\lambda = \tau_{m,\lambda} + AOD_\lambda + \tau_{g,\lambda} \qquad \text{eq. 3}$$

Where  $\tau_{m,\lambda}$ , $\tau_{g,\lambda}$ represent, respectively, molecular scattering and gas absorption optical depths.  All
MFRSR channels are affected by molecular scattering, while gas absorption is highly selective, therefore
affects specific channels. The most relevant influence of gas absorption on MFRSR channels is produced
by Ozone ($O_3$) at 610 and 670 nm channels and by Nitrogen Dioxide ($NO_2$) at 415 nm channel. Therefore,
combination of the eq. 3 and eq. 2 leads to the $AOD_\lambda$ retrieval equation

$$AOD_\lambda = -\frac{1}{m} \ln\left[ \frac{I_{DN,\lambda}}{f(d) <I_{o,\lambda}>} \right] - \tau_{m,\lambda} - \frac{m_{O3}}{m}\tau_{O3,\lambda} - \tau_{NO2,\lambda} \qquad \text{eq. 4}$$

where $\tau_{m,\lambda}$ was calculated using the Kasten and Young (1989) formulation as a function of the
climatological surface atmospheric pressure. Given its unique vertical distribution, ozone relative optical
air mass ($m_{O3}$) was estimated separately based on Staehelin et al. (1995). Ozone and dioxide nitrogen
absorption optical depths were obtained considering their spectral cross section absorption and average
column content over the site, taken from the SCanning Imaging Absorption spectroMeter for Atmospheric
CHartographY (SCIAMACHY, Bovensmann et al., 1999) and Ozone Monitoring Instrument (OMI, Levelt
et al., 2006) products, respectively.
In general, the accuracy of the $AOD_\lambda$ inversion is dominated by uncertainty in the extraterrestrial
response calibration ($<I_{o,\lambda}>$) and $I_{DN,\lambda}$ measurements (Michalsky et al., 2002, Alexandrov et al., 2007,
Mazzola et al., 2010). Typically, uncertainties in both terms are at least one order of magnitude greater than
the contributions of the other terms (Mazzola et al., 2010). Considering only the uncertainties in
extraterrestrial response calibration ($\sigma_{<I_{o,\lambda}>}$) and in  $I_{DN,\lambda}$ measurement ($\sigma_{I_{DN,\lambda}}$), an estimate of uncertainty
($\sigma_{AOD_\lambda}$) of the retrieved $AOD_\lambda$ can be evaluated as

$$\sigma_{AOD_\lambda} = \sqrt{\left[ \frac{1}{m}\frac{\sigma_{<I_{o,\lambda}>}}{<I_{o,\lambda}>} \right]^2 + \left[ \frac{1}{m}\frac{\sigma_{I_{DN,\lambda}}}{I_{DN,\lambda}} \right]^2} \qquad \text{eq. 5}$$





where $\sigma_{<I_{o,\lambda}>}$, as described, is based on the standard error of the mean of multiple extraterrestrial responses
obtained from a set of individual Langley plot calibration. Evaluation of the uncertainty in $I_{DN,\lambda}$ is a
challenge given its dependency on multiple factors, i.e., shadow-band adjustment, accuracy of the angular
response and MFRSR positioning regarding misalignment and tilt (Harrison et al., 1994, Alexandrov et al.,
2007). Harrison et al. (1994) estimated MFRSR $I_{DN,\lambda}$ typical uncertainty to vary between 2 and 3%.
Alexandrov et al. (2007) achieved lower estimation, roughly 1.5% for all channels. Assuming Harrison et
al. (1994) maximum uncertainty (3%), the final uncertainty in MFRSR $AOD_\lambda$, for all channels, was
evaluated for the worst case scenario, i.e., for unit relative air mass($m = 1$).
Additionally, considering $AOD_\lambda$ at two spectral channels ($\lambda1, \lambda2$) as reference, the spectral
dependence of $AOD_\lambda$ was evaluated using Ångström exponent ($\alpha_{\lambda1,\lambda2}$), a parameter inversely related to
the average size of aerosol particles, calculated using the following equation

$$\alpha_{\lambda1,\lambda2} = -\frac{\ln[AOD_{\lambda1}/AOD_{\lambda2}]}{\ln(\lambda1/\lambda2)} \qquad \text{eq. 6}$$

Due to its dependency on aerosol size (Eck et al., 1999), $\alpha_{\lambda1,\lambda2}$ is a practical parameter to evaluate aerosol
particles size. High values of $\alpha_{\lambda1,\lambda2}$, greater than 1.5, indicate dominance of fine aerosol particles, while
values lower than 1.0 and close to zero are typically related to coarse aerosol particles dominance (Eck et
al., 1999). In central Amazonia, for regions up wind of urban areas, such as the T0e site, air masses rich in
fine aerosol particles are typically associated with smoke transport from biomass burning regions. Air
masses dominated by coarse particles fraction are in general associated with local and regional biogenic
and soil particles (Artaxo et al. 1998). Eventually, under favourable atmospheric circulation, air mass
containing coarse dust particles transported from Sahara Desert also may affect T0e site atmospheric
column (Koren et al., 2006, Ben-Ami et al., 2010).
Retrievals of $AOD_\lambda$ and $\alpha_{\lambda1,\lambda2}$ from MFRSR measurements were validated against AERONET
direct Sun products Level 2.0 retrieved by a Cimel sunphotometer also installed at T0e site. AERONET
provides aerosol optical depth at seven wavelengths 340, 380, 440, 500, 670, 870 and 1020 nm, being three



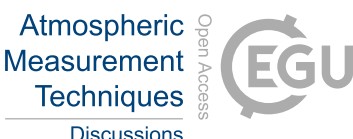

coincident with MFRSR wavelengths, 500, 670 and 870 nm. In order to evaluate the MFRSR $AOD_\lambda$ at the
remaining channels, 415 and 610 nm, the Ångström exponent from AERONET were used to perform
interpolation to derive $AOD_\lambda$ in those channels for the network. Specifically for the comparison purpose,
MFRSR $AOD_\lambda$ at 1 minute rate was averaged within a 5 minute interval centered on AERONET retrieval.
Large standard deviations from the mean were interpreted as cloud contamination in MFRSR, therefore
excluded from the analysis. Afterwards, MFRSR results were used to describe and analyse the seasonal
variability of columnar aerosol particles optical properties over T0e site.
The statistical metrics used to compare MFRSR AOD ($AOD_{MFR}$) with AERONET ($AOD_{Aer}$) , assuming
the later as the true, are  the root mean square error (RMSE), a measure of average deviation from the true,
and Bias, a measure of overall bias error or systematic error:

$$RMSE = \sqrt{\frac{1}{N}\sum_{i=1}^{N}\left(\frac{AOD_{MFR,i} - AOD_{Aer,i}}{AOD_{Aer,i}}\right)^2} \qquad eq.7$$

$$Bias = \frac{1}{N}\sum_{i=1}^{N}\frac{AOD_{MFR,i} - AOD_{Aer,i}}{AOD_{Aer,i}} \quad eq.8$$



## 3. Results

### 3.1 MFRSR Langley plot calibration and uncertainty

An example of a the diurnal cycle of the spectral solar direct-normal irradiance measured (20 June 2012) by the MFRSR prone to a successful Langley plot is presented in **Figure 2**. In the morning period, before vigorous convection initiate, the direct-normal irradiance at all channels is characterized by a continuous increase. The suitability for a successful Langley plot is evidenced in the quality of the linear fit achieved, as can be confirmed in **Table 1** for the 500 nm channel. **Table 1** and **Table 2** present for the 500 nm channels, respectively, for the years 2012 and 2015, the obtained extraterrestrial response calibrations ($I_{o,\lambda}$) for each individual Langley plot that met the criteria defined, i.e. $R^2 \geq 0.990$ and background AOD. The tables with the results for the remain channels (415, 610, 670 and 870 nm) are presented in the supplementary material.

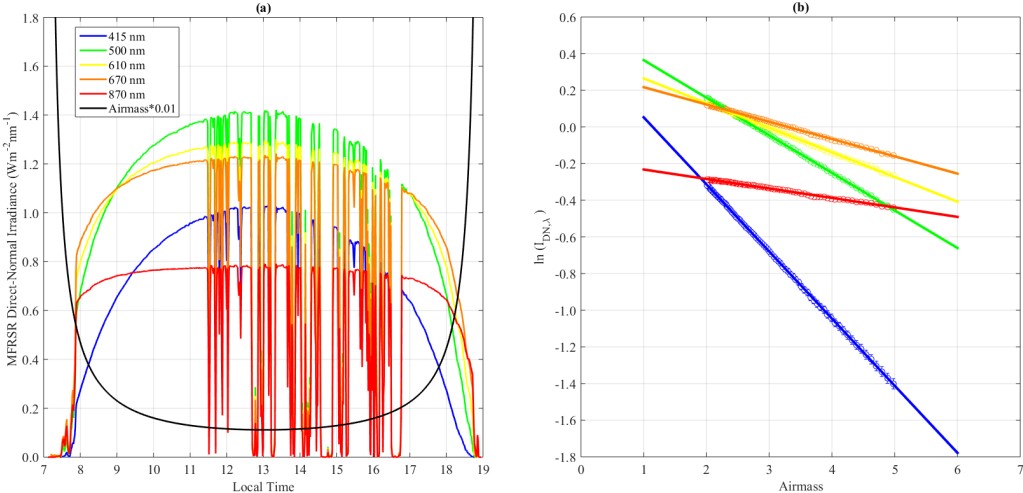

**Figure 2** - (a) Diurnal cycle of air mass and global spectral solar irradiance measured by the MFRSR operating at the T0e site in Central Amazonia. (b) Example of Langley plot calibration applied to MFRSR spectral irradiance measurements taken under the clear sky period (08:00 to 11:00 Local Time) of the diurnal cycle shown in (a). (Day: 20 June 2012)





**Table 1** – Individual extraterrestrial calibration results ($I_{o,500nm}$) applying Langley Plot technique to measurements
of solar direct-normal irradiance at 500 nm from a MFRSR operating at T0e site in Central Amazônia for the year
2012. The individual uncertainty [$\sigma\_Io.\lambda$ ] used to obtain the relative error [$\sigma\_Io.\lambda$ (%)] was estimated from the
**intercept** and its respective uncertainty ($\sigma\_intercept$) derived from the least square regression method.

| Date | slope | $\sigma\_slope$ | intercept | $\sigma\_intercept$ | $I_{o.500\,nm}$ | $\sigma\_Io.\lambda$ (%) | $R^2$ | N |
|---|---|---|---|---|---|---|---|---|
| 17-may-12 | -0.2426 | 0.0016 | 0.5709 | 0.0043 | 1.814 | 0.434 | -0.9992 | 63 |
| 16-jun-12 | -0.2450 | 0.0019 | 0.6058 | 0.0055 | 1.895 | 0.549 | -0.9939 | 64 |
| 17-jun-12 | -0.2237 | 0.0016 | 0.5560 | 0.0046 | 1.803 | 0.464 | -0.9990 | 61 |
| 20-jun-12 | -0.2117 | 0.0015 | 0.5846 | 0.0043 | 1.856 | 0.434 | -0.9992 | 64 |
| 21-jun-12 | -0.2261 | 0.0017 | 0.5722 | 0.0047 | 1.834 | 0.474 | -0.9996 | 65 |
| 22-jun-12 | -0.2265 | 0.0018 | 0.5362 | 0.0050 | 1.769 | 0.501 | -0.9995 | 71 |
| 25-jun-12 | -0.2585 | 0.0019 | 0.6461 | 0.0055 | 1.975 | 0.546 | -0.9992 | 78 |
| 3-jul-12 | -0.2493 | 0.0020 | 0.5848 | 0.0058 | 1.858 | 0.577 | -0.9978 | 61 |
| 4-jul-12 | -0.2436 | 0.0019 | 0.6060 | 0.0054 | 1.898 | 0.542 | -0.9998 | 63 |
| 8-jul-12 | -0.2430 | 0.0020 | 0.5668 | 0.0058 | 1.824 | 0.581 | -0.9996 | 64 |
| 11-jul-12 | -0.2420 | 0.0021 | 0.5456 | 0.0059 | 1.785 | 0.590 | -0.9995 | 62 |
| 1-aug-12 | -0.2616 | 0.0021 | 0.5843 | 0.0058 | 1.848 | 0.580 | -0.9997 | 64 |
| 2-aug-12 | -0.2401 | 0.0020 | 0.5221 | 0.0055 | 1.736 | 0.549 | -0.9920 | 62 |
| 3-aug-12 | -0.2775 | 0.0021 | 0.6313 | 0.0058 | 1.935 | 0.584 | -0.9912 | 65 |
| 4-aug-12 | -0.2359 | 0.0017 | 0.5751 | 0.0048 | 1.829 | 0.482 | -0.9991 | 62 |
| 6-aug-12 | -0.2880 | 0.0025 | 0.5561 | 0.0070 | 1.793 | 0.700 | -0.9987 | 63 |
| 21-dec-12 | -0.2658 | 0.0016 | 0.6294 | 0.0042 | 1.815 | 0.418 | -0.9996 | 63 |

**Table 2** – Individual extraterrestrial calibration results ($I_{o,500nm}$) applying Langley Plot technique to measurements
of solar direct-normal irradiance at 500 nm from a MFRSR operating at T0e site in Central Amazônia for the year
2015. The individual uncertainty [$\sigma\_Io.\lambda$ ] used to obtain the relative error [$\sigma\_Io.\lambda$ (%)] was estimated from the
**intercept** and its respective uncertainty ($\sigma\_intercept$) derived from the least square regression method.

| Date | slope | $\sigma\_slope$ | intercept | $\sigma\_intercept$ | $I_{o.500\,nm}$ | $\sigma\_Io.\lambda$ (%) | $R^2$ | N |
|---|---|---|---|---|---|---|---|---|
| 19-feb-15 | -0.2045 | 0.0014 | 0.5723 | 0.0041 | 1.734 | 0.412 | -0.9959 | 62 |
| 27-mar-15 | -0.2335 | 0.0015 | 0.5957 | 0.0039 | 1.809 | 0.395 | -0.9941 | 69 |
| 4-jun-15 | -0.2787 | 0.0021 | 0.6436 | 0.0058 | 1.963 | 0.583 | -0.9923 | 68 |



| 24-jun-15 | -0.1900 | 0.0013 | 0.5545 | 0.0039 | 1.802 | 0.394 | -0.9996 | 63 |
|---|---|---|---|---|---|---|---|---|
| 1-jul-15 | -0.2301 | 0.0016 | 0.6247 | 0.0048 | 1.933 | 0.478 | -0.9989 | 62 |
| 2-jul-15 | -0.2039 | 0.0015 | 0.5530 | 0.0043 | 1.800 | 0.433 | -0.9995 | 62 |
| 6-jul-15 | -0.2397 | 0.0019 | 0.6022 | 0.0054 | 1.890 | 0.542 | -0.9979 | 61 |
| 10-jul-15 | -0.2513 | 0.0019 | 0.6256 | 0.0055 | 1.934 | 0.546 | -0.9988 | 61 |
| 11-jul-15 | -0.2487 | 0.0019 | 0.6169 | 0.0056 | 1.917 | 0.556 | -0.9996 | 61 |
| 12-jul-15 | -0.2634 | 0.0022 | 0.5949 | 0.0063 | 1.876 | 0.634 | -0.9993 | 61 |
| **15-jul-15** | -0.2896 | 0.0026 | 0.6070 | 0.0074 | 1.898 | 0.745 | -0.9994 | 61 |
| 28-jul-15 | -0.2606 | 0.0020 | 0.6344 | 0.0056 | 1.945 | 0.555 | -0.9982 | 62 |
| 29-jul-15 | -0.2496 | 0.0021 | 0.5611 | 0.0059 | 1.807 | 0.585 | -0.9901 | 62 |
| 30-jul-15 | -0.2406 | 0.0018 | 0.5912 | 0.0051 | 1.862 | 0.510 | -0.9964 | 62 |
| 1-aug-15 | -0.2500 | 0.0019 | 0.6162 | 0.0054 | 1.908 | 0.536 | -0.9954 | 62 |
| 2-aug-15 | -0.2907 | 0.0024 | 0.6385 | 0.0066 | 1.950 | 0.657 | -0.9983 | 62 |
| 7-aug-15 | -0.2535 | 0.0018 | 0.6151 | 0.0051 | 1.902 | 0.508 | -0.9997 | 64 |
| 23-aug-15 | -0.2652 | 0.0018 | 0.6047 | 0.0048 | 1.870 | 0.482 | -0.9987 | 69 |
| **5-sep-15** | -0.2623 | 0.0018 | 0.5373 | 0.0044 | 1.737 | 0.438 | -0.9983 | 74 |
| 9-sep-15 | -0.2411 | 0.0014 | 0.6266 | 0.0038 | 1.895 | 0.376 | -0.9996 | 75 |
| 22-sep-15 | -0.2825 | 0.0018 | 0.5998 | 0.0045 | 1.831 | 0.454 | -0.9992 | 75 |

The final extraterrestrial response estimations $< I_{o,\lambda} >$, for both years and all channels, based on average
of all individual Langley plot calibration, are presented in **Table 3** along the standard error from the mean
as the uncertainty $(\sigma_{<I_{o,\lambda}>})$, sample number (N) and the relative difference between calibration estimate
for 2012 and 2015.  The relative uncertainties among the channels varied from 0.7% (870 nm) to 1.0% (415
nm) in 2012, and from 0.4% (870 nm) to 1.0% (415 nm) in 2015, which are surprisingly satisfactory for
conditions diverse from those recommended (clean top mountain). Regarding the relative difference (-
0.4%) between calibration constant derived for the two years, the difference for the channel 415 nm is not
statistically significant, suggesting that between 2012 and 2015 the correspondent transmission filter did
not suffer relevant degradation. Meanwhile, a drift of 4.8 % was observed for the 870 nm channel, an





indication of the lower stability of its transmission filter. The remain channels (500, 613, 670 nm)
calibrations constant, opposite to the 870 nm channel, presented positive trend between 2012 and 2015
calibrations. However, given the values of the uncertainty $(\sigma_{<I_{o,\lambda}>})$ in their calibration constants, we are
not able to attest that 500, 613 and 670 nm channels have statistically suffered degradation.
**Table 3** – MFRSR final extraterrestrial calibrations estimates $< I_{o,\lambda} >$ for the years 2012 and 2015 averaging results
of individual Langley plot calibration from Table 1, Table 2 and Tables in the supplementary material. The
uncertainty estimation $(\sigma_{<Io,\lambda>})$ is based on the correspondent standard error of the average.

| Channels | \multicolumn{3}{c}{Year 2012} | | | \multicolumn{3}{c}{Year 2015} | | | Difference (%) |
|---|---|---|---|---|---|---|---|
| | N | $<I_o>$ | $\sigma_{<I_{o,\lambda}>}$ | N | $<I_o>$ | $\sigma_{<I_{o,\lambda}>}$ | $\Delta<I_{o,\lambda}>$ (2012-2015) |
| 415 nm | 21 | 1.586 | 0.015 (1.0 %) | 22 | 1.579 | 0.017 (1.0%) | -0.4 |
| 500 nm | 17 | 1.839 | 0.015 (0.8 %) | 21 | 1.870 | 0.015 (0.8%) | +1.7 |
| 613 nm | 14 | 1.545 | 0.010 (0.7%) | 17 | 1.572 | 0.011 (0.7%) | +1.8 |
| 670 nm | 15 | 1.416 | 0.010 (0.7%) | 18 | 1.433 | 0.008 (0.6%) | +1.2 |
| 870 nm | 15 | 0,842 | 0.008 (0.9%) | 20 | 0.802 | 0.003 (0.4%) | -4.8 |

Considering the estimate uncertainties in the extraterrestrial calibration constant (0.4% -1.0%), and
the Harrison et al. (1994) maximum uncertainty (3%) for MFRSR $I_{DN,\lambda}$ measurements, accordingly to the
error propagation analysis (equation 6), the worst estimative (i.e., for unit airmass) for our absolute
uncertainty in $AOD_\lambda$ is ~ 0.03, which is comparable with uncertainty of $AOD_\lambda$ retrieved from AERONET
field sunphotometers measurements (~0.02, Eck et al., 1999). However, if a lower uncertainty in $I_{DN,\lambda}$ is
assumed, for instance 1.5% (as suggested by Alexandrov et al., 2007), that would reduce MFRSR $AOD_\lambda$
uncertainty from ~0.03 to ~ 0.02.
In the general, perfect linear Langley plots are associated with stable aerosol optical depth, however
it is possible that not all nearly linear Langley plots are able to provide correct calibration. Airmass
assumption, mainly regarding aerosol particles airmass (Schmid and Wehrli, 1995), instruments induced
artefact, the shadow-band system alignment (Chen et al., 2013), may contribute to error in calibration.



These influences are all challenge to estimate. Therefore, taking the mean of a set of individual Langley
plot calibration as the best estimate for the final calibration constant along the comparison of the AOD
results with AERONET retrieval should provide a good reference to evaluate the quality of the calibration
constant obtained. The results obtained for RMSEs derived from the comparison between MFRSR
retrievals and AERONET AOD are lower than the estimated uncertainty for MFRSR $AOD_\lambda$ retrievals (i.e.,
~0.02 - 0.03, depending on the $I\_(DN,\lambda)$ uncertainty assumed (1.5 or 3 %) and just above the maximum
uncertainty for AERONET field instrument (~0.02), demonstrating that, in spite of eventual  error
associated with assumption made during the Langley plot application, the final derived constants are able
to provide reliable AOD retrievals.
**3.2 Aerosol Optical Depth $(AOD_\lambda)$ inversion and uncertainty estimate**

12   Once determined the MFRSR channels final extraterrestrial response calibration, direct-normal

irradiance measurements taken along 2012 and 2015 were applied to retrieve $AOD_\lambda$ and calculate
Ångström exponent. **Figure 3** illustrates, for a specific day (22 November 2012), results of cloud screening
and a comparison between the diurnal variability of $AOD_\lambda$ from MFRSR and AERONET. The cloud
screening criteria captured the majority of contaminated measurements, but few suspicious remaining
points are likely related to optically thin cirrus. A more conservative algorithm would remove a significant
amount of cloud free cases, as seems to be the case for AERONET retrievals. The intercomparison showed
the consistency of MFRSR retrievals regarding $AOD_\lambda$ diurnal variability. It is worth to emphasize the higher
frequency of MFRSR retrieval during the afternoon when compared with AERONET product. This is a
critical aspect regarding the representativity of $AOD_\lambda$ diurnal variation in regions marked by strong diurnal
cycle of convection and cloud cover such as Central Amazonia. The MFRSR one minutes frequency is
expected to improve the statistic of AOD under cloudy conditions, since AERONET current statistics are
recognized to be biased toward cloudless sky conditions (Levy et al, 2010).



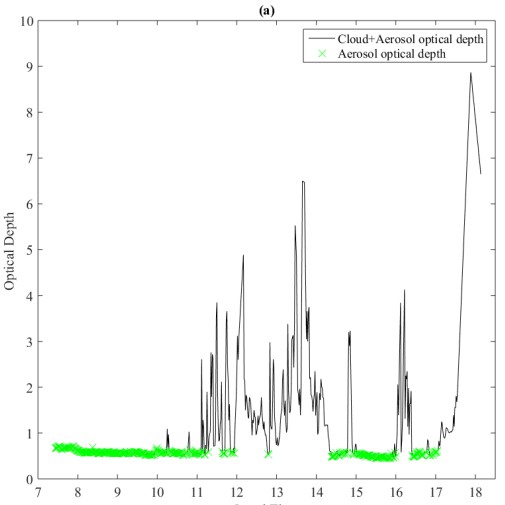
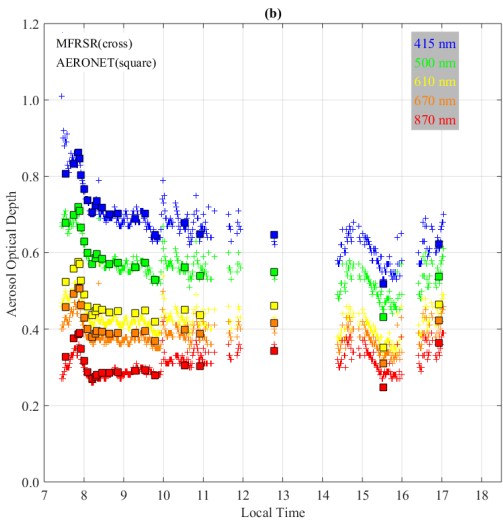

**Figure 3** - (a) Example of the cloud screening applied to the MFRSR optical depth retrievals (22 November 2012). (b) Cloud screened diurnal cycle of multichannel aerosol optical depth from MFRSR compared with AOD retrievals from AERONET Level 2.0 product.

A comparison focusing on seasonal variability was also performed. **Figure 4** presents the 2012 seasonal variability of $AOD_{500\,nm}$ and $\alpha_{415,670\,nm}$ over T0e site as seen by MFRSR (based on 1 min time resolution) and AERONET. MFRSR retrievals were able to represent consistently the major seasonal features. From March to June, central Amazonia presents its lowest $AOD_{500\,nm}$ levels, ranging from ~0.05 to ~0.20. In a completely opposite scenario, during the biomass burning season (August to November), $AOD_{500\,nm}$ hardly goes down below 0.20 and values above 0.50 are quite frequent. During the transition periods, from background conditions to biomass burning (June to July) and from biomass burning to background (December to February), $AOD_{500\,nm}$ values oscillated between typical background and biomass burning season. Considering that the enhancement of $AOD_\lambda$ during the biomass burning season across central Amazonia is dominated by increase in small particles (Eck et al., 1999, Rosario, 2011), $\alpha_{415,670\,nm}$ variability (**Figure 4**) is consistent with the $AOD_{500}$ discussion, i. e., as the aerosol loading increase from July to the biomass burning month (Aug-Sep-Oct-Nov), $\alpha_{415,670\,nm}$ also sows and enhancement. Ångström Exponents ranging from 0.4 to 0.8, which are dominant under background conditions, became



rare throughout the biomass burning season and intermittent during the transition periods, a feature
consistently described by MFRSR and AERONET. Similar results, for both $AOD_{500}$ and $\alpha_{415,670\,nm}$
were observed regarding the year 2015 (not shown here).

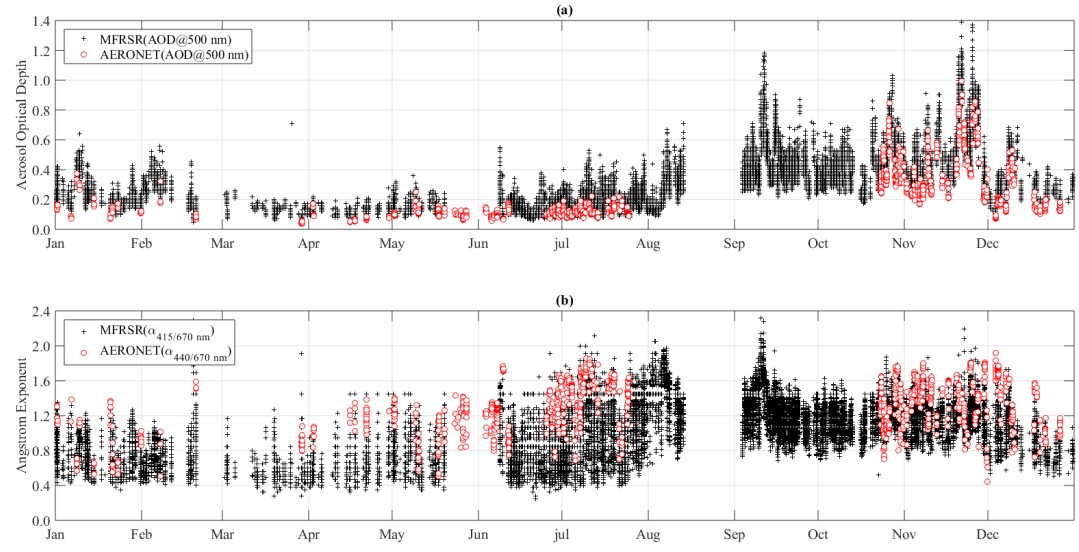

**Figure 4** – (a) Seasonal aerosol optical Depth and (b) Angstrom exponent variability in Central Amazonia as obtained via the on-site calibrated MFRSR (this study) and the AERONET sunphotometer for the year 2012.

**Figures 5** and **6** show scatter plots and statistic metrics (Bias, RMSE and Correlation coefficient)
comparing MFRSR and AERONET retrievals for 2012 and 2015, respectively. In general, there is a good
agreement between both $AOD_\lambda$ retrievals. However, non-negligible trends are seen, especially for 2012,
and in particular for the lower and higher AOD edges. For low $AOD_\lambda$ values, a systematic underestimation
by MFRSR is observed for all channels, while for high $AOD_\lambda$, the longer wavelength channels (610 and
670 nm) tend to underestimate aerosol optical depth. For the 2015 years trends are less evident, mainly for
the low aerosol loading when compared with 2012. Nevertheless, overall, the statistics metrics (**Table 4**)
used to evaluate MFRSR retrievals performance against AERONET suggest that, when is not possible to
perform high top mountain calibration, the extraterrestrial response calibration performed at Central





Amazonia has reliability to support consistent retrievals of aerosol optical depth. The obtained RMSEs are
lower than the estimated uncertainty for MFRSR $AOD_\lambda$ retrievals (i.e., ~0.02 - 0.03, depending on the $I_{DN,\lambda}$
uncertainty assumed) and just above the maximum uncertainty for AERONET field instrument (~0.02).

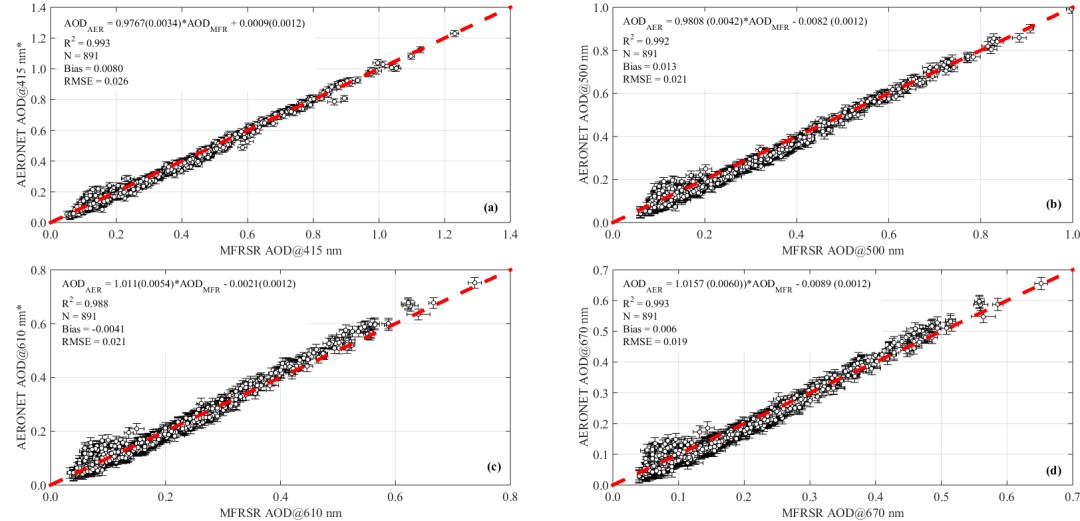

**Figure 5** - Spectral AOD retrieval from the on-site calibrated MFRSR as a function of AOD from
AERONET direct product level 2.0 for the year 2012.

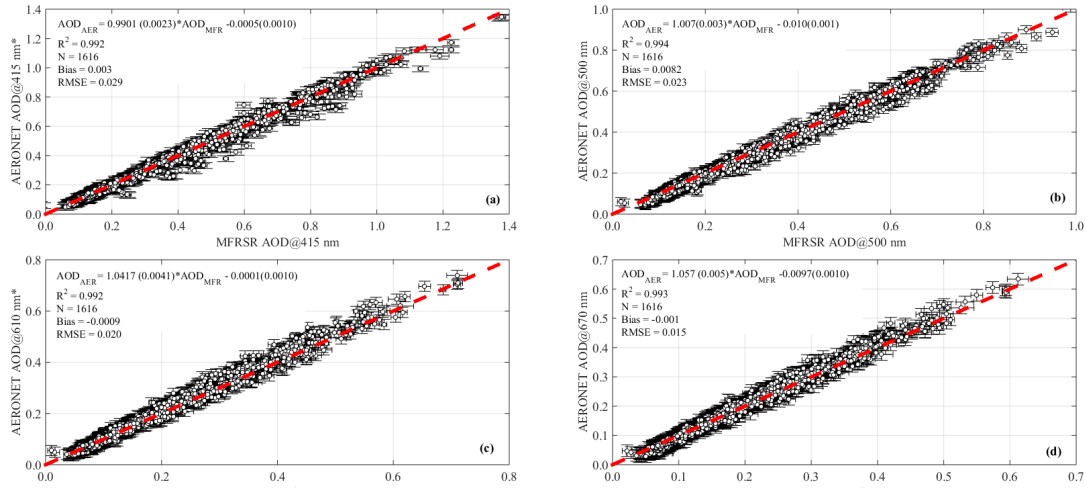

**Figure 6** - Spectral AOD retrieval from the on-site calibrated MFRSR as a function of AOD from
AERONET direct product level 2.0 for the year 2015.





**Table 4** – Summary of the statistical metrics used to evaluate MFRSR aerosol optical depth (X) against AERONET
optical depth (Y) for the coincident spectral channels (500, 670 and 870 nm): Bias, Root Mean Square Error (RMSE).
The linear fit regression parameters, slope (A) and intercept (B) and correspondent uncertainty ($\sigma_A$; $\sigma_B$) are also
presented.

|      | $\lambda$ | N | Bias | RMSE | Y = A ($\sigma_A$)*X + B ($\sigma_B$) | $R^2$ |
|------|-----------|---|------|------|----------------------------------------|-------|
| **2012** | 500 nm | 891 | 0.0130 | 0.021 | A = 0.9808 (0.0042); B = -0.0082 (0.0012) | 0.992 |
|      | 670 nm | 891 | 0.0064 | 0.019 | A = 1.0157 (0.0060); B = -0.0089 (0.0012) | 0.988 |
|      | 870 nm | 891 | 0.0148 | 0.031 | A = 0.901 (0.007); B = -0.0016 (0.0012) | 0.962 |
| **2015** | 500 nm | 1616 | 0.0082 | 0.023 | A = 1.007 (0.003); B = -0.010 (0.001) | 0.994 |
|      | 670 nm | 1616 | -0.0010 | 0.015 | A = 1.057 (0.005); B = -0.0097 (0.0010) | 0.993 |
|      | 870 nm | 1616 | -0.0040 | 0.016 | A = 1.1285 (0.0073); B = -0.0136 (0.0011) | 0.9895 |

## 4. Conclusions
Do Central Amazonian pristine atmosphere provides successful extraterrestrial response calibration based
on Langley plot method? This question emerged from the challenge to maintain regular calibration of a
MFRSR dedicated to long-term retrieval of columnar aerosol optical properties in central Amazônia. To
answer the question, the MFRSR was calibrated on site using the Langley plot method for two distinct
years, 2012 and 2015, and subsequently applied to retrieve aerosol columnar optical properties, i.e., Aerosol
Optical Depth (AOD) and Ångström Exponent (AE). Retrievals were evaluated against direct sun inversion
products (Level 2.0) from a collocated sunphotometer belonging to AERONET. Results obtained show that
on site calibration using Langley plot, under Amazonian pristine conditions, is able to provide
extraterrestrial response with relative uncertainties varying from ~0.4 to ~1.0 % at MFRSR visible channels.
The worst estimative (airmass = 1) for absolute uncertainty in retrieved $AOD_\lambda$ can varied from ~0.03 to
~0.02, depending on the assumption regarding the uncertainty assumed for MFRSR direct-normal
irradiance measured at the surface $\left( I_{DN,\lambda} \right)$, which in the literature varied from 1.5% to 3.0%. All Root
Mean Square Error (RMSE), obtained from the comparison of MFRSR retrievals against AERONET $AOD_\lambda$
for coincident channels (500 and 670 nm), were lower (< 0.025) than the estimated MFRSR $AOD_\lambda$





uncertainties and close to AERONET field sunphotometers (~ 0.02). Under the point of view of the question
posed, these results suggest that on site calibration in central Amazonia pristine conditions is able to provide
consistent retrieval of $AOD_\lambda$. Another relevant aspect of the results provided by the MFRSR, due to its high
measurement frequency (one minute), is the improvement of the statistic of AOD under cloudy conditions,
which is critical for Amazonia. AERONET current statistics are expected to be biased to cloudless sky
conditions, which are dominant during the morning period and dry season.
**Competing interests**. The authors declare that there are no competing interests
**Acknowledgements.** The authors would like to EMBRAPA, INPA, and the LBA Central office for
logistical support. Special thanks to Marcelo Rossi, Victor Souza, and Jocivaldo Souza at EMBRAPA, and
to Ruth Araujo, Roberta Souza, Bruno Takeshi, and Glauber Cirino from LBA. Henrique Barbosa
acknowledges the financial support from FAPESP Research Program on Global Climate Change under
research grants 2008/58100-1, 2012/16100-1, 2013/50510-5, and 2013/05014-0. Theotonio Pauliquevis
acknowledges the financial support from CNPq research grant 458017/2013-2. Boris Barja acknowledges
the financial support of CAPES project A016_2013 on the program Science without Frontiers and the
SAVERNET project.





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
