# Peer review of "Aerosol Optical Depth retrievals in Central Amazonia from a Multi- Filter Rotating Shadow-band Radiometer on-site calibrated"

_Atmospheric Measurement Techniques, 2018_

## Referee Comment (RC1) · J. Michalsky (Referee) · 11 Jun 2018

The authors main goal was to determine whether it was feasible to obtain a meaningful calibration of a sun radiometer in a less than optimal locale for performing Langley calibrations. Langley calibrations allow one to estimate the TOA response of a sun radiometer, but are best performed on a high mountain top above the boundary layer. To this end they compared aerosol optical depths obtain from an MFRSR with the CIMEL radiometer operated using the AERONET protocol. The CIMEL calibration is derived from comparison to instruments calibrated at Mauna Loa Observatory. The RMSE, which they define as a deviation from the AERONET results, was 0.025 and

within the uncertainties of the two instruments. I find that the results of the paper are based on scientifically sound reasoning and should be acceptable for publication.

Should the authors be so inclined, I am curious whether the results would change if some other estimate of Vo's such at the median or the method used in Michalsky et al. (2001) had been used to obtain Vo's.

Could the authors explain why 2013 and 2014 data were not included?

A plot of Vo's might be helpful in demonstrating the stability of the Langley results in most of the filters with the 870-nm filter an exception. It would also perhaps demonstrate the lack of a seasonal dependence seen in other MFRSRs since the temperature of the central Amazon is rather stable throughout the year.

There are a few grammatical and spelling errors, but none so egregious as to make the text misunderstood.

---

## Referee Comment (RC2) · Anonymous Referee #2 · 5 Aug 2018

This article has as main goal of establishing and verifying an alternative setup for MFRSR calibration over the Amazonian basin. The need is clear: Amazonian atmosphere have to be continuously monitored and gaps (for example to send the instrument to a calibration facility) must be avoided. I think the authors achieved their objective with this work. Moreover, this article present a good example of comparison with the AERONET network, in terms of aerosol optical depth and Angström exponent. I find that the paper will be scientifically sound and it might be acceptable for publication after addressing the major points listed in my specific comments.

Specific comments:

[Figure]

Page 3, lines 12-14: For non-familiarized reader it would be more convenient to describe on Figure 1 the meaning of midst, upwind and downwind pollution plume. Is the plume generated inside Manaus city? Which kind of particles are mainly present?

Page 8, lines 12-16: Ozone and dioxide nitrogen content was obtained from Sciamachy and OMI, but did you use a daily value, a monthly value or an average value over 2012-2015?

Page 9, line 2: How many individual Langley calibrations were performed over the period 2012-2015? Is this number robust enough? Only information on years 2012 and 2015 is included in the manuscript and supplement material.

Page 9, lines 8-9: You wrote "$\alpha\lambda1,\lambda2$ is a practical parameter to evaluate aerosol particles size". This sentence is too general. What it can be inferred from the Angstöm exponent is the predominance of submicrometric or micrometric particles, but not the actual particle size. For that, information on particle size distribution (for instance) must be retrieved.

Page 9, line21: please include also the reference Moran-Zuloaga et al. (2018)

Moran-Zuloaga, D., Ditas, F., Walter, D., Saturno, J., Brito, J., Carbone, S., Chi, X., Hrabě de Angelis, I., Baars, H., Godoi, R. H. M., Heese, B., Holanda, B. A., Lavrič, J. V., Martin, S. T., Ming, J., Pöhlker, M. L., Ruckteschler, N., Su, H., Wang, Y., Wang, Q., Wang, Z., Weber, B., Wolff, S., Artaxo, P., Pöschl, U., Andreae, M. O., and Pöhlker, C.: Long-term study on coarse mode aerosols in the Amazon rain forest with the frequent intrusion of Saharan dust plumes, Atmos. Chem. Phys., 18, 10055-10088, https://doi.org/10.5194/acp-18-10055-2018, 2018.

Page 9, line 24: replace "aerosol optical depth" by "AOD" because you have already introduced the acronym. Check along the manuscript.

Page 10, lines 5-6: Cloud screening is done based on the large standard deviations from the mean observed in some cases. However, it is necessary to quantify what you

refer with "large standard deviations".

Page 10, lines 8-9: replace the word "true" by "reference". Page 12, Table 1 and Table 2: Following equation 1 the slope (after changing its sign) provides information on the daily average of AOD. All the values shown in Table 1 and Table 2, both referred to 500 nm, are in the range 0.20-0.30, what from my point of view are not low enough to be considered as clear atmosphere or values similar to those obtained at a mountain top (above the atmospheric boundary layer). In this sense, more discussion is needed.

Page 14, line 2 (but also this is an overall comment): Only years 2012 and 2015 are analyzed in this paper? What about 2013 and 2014? Is there any reason for this lack of information?

Page 15, line 17: You attributed the few suspicious points to the presence of optically thin cirrus. This can be easily checked from lidar measurements. Do you have simultaneous lidar measurements to corroborate this?

Page 16, line 5: AERONET is not an instrument, replace by Sun-photometer or the AERONET Sun-photometer. The same in page 7, line 2.

Page 17, line 3: what about the results for 2013 and 2014? Page 17, figure 4: From this figure it seems that there is a overestimation of AOD from MFRSR respect to Cimel values, and underestimation of Angström Exponent values. Due to the different temporal resolutions I consider more convenient to present the temporal series of daily values or monthly values to check the overall coherence of both datasets. If monthly values are shown, the whole dataset (2012-2015) can be presented.

Page 18, figure 5: All wavelengths should be shown here. Also, explain the meaning of asterisk on the y-axis unit (interpolated values, I guess), and the meaning of red dotted line (1:1 line).

Page 18, figure 6: The information from Figure 6 is summarized in Table 6. It would be nice if you replace figure 6 about AOD by the scatter plot of Angström exponents.

Page 19, Table 4: All wavelengths and years must be shown here. Other table 7 might contain the corresponding information for Angström exponent

Technical corrections:

Page 2, line 26: in "...the forest it is..." remove "it".

Page 7, line 7: replace "...using leas-square..." by "...using least-square...".

Page 15, line6: keep the same format along the paper (don't use I_(DN,$\lambda$) ).

Page 15, line 22: replace "one minutes frequency" by "1-min frequency".

Page 16, line 14: replace "sows and enhancement" by "shows an enhancement".

Page 17, caption figure 4: replace "Depth" by "depth".

Page 17, line 11: replace "For the 2015 years trends" by "The year 2015 trends".

Page 19, Table 4, caption: replace "aerosol optical depth" by "AOD" and "optical depth" by "AOD".

Page 19, line 7: replace "Do Central Amazonian pristine atmosphere provides" by "Does Central Amazonian pristine atmosphere provide".

Page 19, line 9: "Amazônia" Please, in English not in Portuguese.

Page 19, line 18: replace "varied" by "varies".

―――――――――――――――――

---

## Author Comment (AC1) · 14 Sep 2018

**joseph.michalsky@noaa.gov**

*"The authors main goal was to determine whether it was feasible to obtain a meaningful calibration of a sun radiometer in a less than optimal locale for performing Langley calibrations. Langley calibrations allow one to estimate the TOA response of a sun radiometer, but are best performed on a high mountain top above the boundary layer. To this end they compared aerosol optical depths obtain from an MFRSR with the CIMEL radiometer operated using the AERONET protocol. The CIMEL calibration is derived from comparison to instruments calibrated at Mauna Loa Observatory. The RMSE, which they define as a deviation from the AERONET results, was 0.025 and within the uncertainties of the two instruments. I find that the results of the paper are based on scientifically sound reasoning and should be acceptable for publication."*

**Authors general comments**: We are glad that the manuscript content was appreciated and we would like to thank the referee for the interesting points highlighted. We have tried to address the points raised. Below we provide answers to each of your comments.

Legend:
**Q#<number>** - Referee questions and suggestion
**R#<number>** - Authors reply and comments

**Q#01:"** Should the authors be so inclined, I am curious whether the results would change if some other estimate of Vo's such at the median or the method used in Michalsky et al. (2001) had been used to obtain Vo's."

**R#01**: We used median to estimate Vo's for both years 2012 and 2015 (Tables below). The results agree within ~1% for all wavelengths. In general, medians presented slightly higher values, except for 500 and 610 nm in 2012.

| Year 2012 | 415 nm | 500 nm | 610 nm | 670 nm |
|---|---|---|---|---|
| Mean | 1.586 ±0.015 (1%) | 1,839±0.015 (0.8%) | 1.545±0.015 (0.7%) | 1.416±0.015 (0.7%) |
| Median | 1.586 | 1.829 | 1.537 | 1.405 |
| Median–Mean (%) | 0.001 (0.1%) | -0.010 (0.6%) | -0.008 (0.5%) | -0.011 (0.7%) |

| Year 2015 | 415 nm | 500 nm | 610 nm | 670 nm |
|---|---|---|---|---|
| Mean | 1.579±0.017 (%1.1) | 1.870±0.015 (0.8%) | 1.572±0.011(0.7%) | 1.433±0.008 (0.6%) |
| Median | 1.582 | 1.890 | 1.592 | 1.443 |
| Median-Mean | 0.0035 (0.2%) | 0.020(1.1%) | 0.019(1.2%) | 0.010(0.7%) |

**Q#02**:" Could the authors explain why 2013 and 2014 data were not included?"

**R#02:** As we set the focus of the manuscript on the question whether it is possible to obtain accurate calibration constants derived from on site measurements applying the Langley plot method in Central Amazonia, we evaluated that two independent years would be adequate to support our findings concerning the question. That is the main reason why we present only 2012 and 2015. We selected 2012 and 2015 because of the temporal distance between them, which would allow us to detect a scenario of potential filter degradations. Now that we evaluated that consistent AOD retrievals, derived from local successful calibration constants, can be obtained, there is an ongoing study focusing on a multi-year analysis of AOD. We plan to include a broad discussion in terms of source contributions and atmospheric processes and also a time series of the calibration constant applied to obtain the correspondent MFRSR AOD values.

**Q#03:**" A plot of Vo's might be helpful in demonstrating the stability of the Langley results in most of the filters with the 870-nm filter an exception. It would also perhaps demonstrate the lack of a seasonal dependence seen in other MFRSRs since the temperature of the central Amazon is rather stable throughout the year."

**R#03:** A challenge that we faced in the attempt to evaluate the lack of a seasonal dependence is that during the wet season, when Amazon is too cloudy, we were not able to obtain a significant number of Langley plot, most of the Langley plot are at the beginning and in the middle of the dry season.

**Q#04:**" There are a few grammatical and spelling errors, but none so egregious as to make the text misunderstood"

**R#04:** We went throughout the text and tried to identify and correct all remaining grammatical and spelling errors.

---

## Author Comment (AC2) · 14 Sep 2018

This article has as main goal of establishing and verifying an alternative setup for MFRSR calibration over the Amazonian basin. The need is clear: Amazonian atmosphere have to be continuously monitored and gaps (for example to send the instrument to a calibration facility) must be avoided. I think the authors achieved their objective with this work. Moreover, this article present a good example of comparison with the AERONET network, in terms of aerosol optical depth and Angström exponent. I find that the paper will be scientifically sound and it might be acceptable for publication after addressing the major points listed in my specific comments.

**Authors general comments**: We would like to thank the referee for his thoughtful revision and suggestions. We have tried to address all points raised, that certainly will improve our manuscript. Below we provide answers to each of your comments.

Legend:
**Q#<number>** - Referee questions and suggestion
**R#<number>** - Authors reply and comments

**Specific comments:**
**Q#01:** Page 3, lines 12-14: For non-familiarized reader it would be more convenient to describe on Figure 1 the meaning of midst, upwind and downwind pollution plume. Is the plume generated inside Manaus city? Which kind of particles are mainly present?
**R#01**: Figure 1 was updated, we included more information: wind dominant direction during the dry (brown arrow) and wet (blue arrow) seasons. During the wet season (November – March) the dominant wind direction is from northeast and during the dry from east. We added examples of the GoAmazon sites to reference the downwind (T2), midst (T1) and upwind (T0e) position respect to the Manaus city, being the later site (T0e) the focus of the present study.

[Figure]

Regarding the composition of Manaus plume, nitrogen and sulphur oxides, submicron aerosol particles and soot are found in high concentrations (Kuhn et al., 2010), which is consistent with the nature of the major sources of air pollution from Manaus, vehicle fleet,

power plants, and industrial activities. Sá et al. (2017) found that the submicron particles composition is dominated by organic material across measurement sites upwind and downwind of Manaus, independently of the levels of pollution. However, their study pointed out that, among the sites, the absolute mass concentrations varied significantly. Average concentrations downwind of Manaus were 100 % to 200 % higher than those upwind.

This description was included in the new version of the manuscript that is being prepared

**Q#02:** Page 8, lines 12-16: Ozone and dioxide nitrogen content was obtained from Sciamachy and OMI, but did you use a daily value, a monthly value or an average value over 2012-2015?

**R#02:** For both Ozone (268 Dobson Units) and NO2 (7.6 Dobson Units) column content, we used average values over 2011 - 2015. This information was included in text.

**Q#03:** Page 9, line 2: How many individual Langley calibrations were performed over the period 2012-2015? Is this number robust enough? Only information on years 2012 and 2015 is included in the manuscript and supplement material.

**R#03:** As we set the focus of the manuscript on the question whether it is possible to obtain accurate calibration constants derived from on site measurements applying the Langley plot method in Central Amazonia, we evaluated that two independent years would be adequate to support our findings concerning the question. That is the main reason why we present only 2012 and 2015. We selected 2012 and 2015 because of the temporal distance between them, which would allow us to detect a scenario of potential filter degradations. Now that we evaluated that consistent AOD retrievals, derived from local successful calibration constants, can be obtained, there is an ongoing study focusing on a multi-year analysis of AOD. We plan to include a broad discussion in terms of source contributions and atmospheric processes and also a time series of the calibration constant applied to obtain the correspondent MFRSR AOD values.

Regarding the number of Langley calibrations (that varied from 14 to 22, depending on the year and wavelength channel), we think that those figures allow a consistent statistic for calibration constants. The numbers are similar to those of previous and references studies on the Langley plot calibration, for instance, Schmid and Wehrli ,1995 (11 Langley plot cases) Michalsky et al., 2001 (20 Langley plot cases), Augustine et al., 2003 (18 Langley plot cases). Another important aspect to corroborate the quality of the Langley plots performed is that, in our study, we included more than 60 points per Langley plot, when 20 is suggested as a minimum to obtained good results from an individual Langley plot (Augustine et al., 2003).

**Q#04:** Page 9, lines 8-9: You wrote "$\alpha\lambda 1, \lambda 2$ is a practical parameter to evaluate aerosol particles size". This sentence is too general. What it can be inferred from the Angstöm exponent is the predominance of submicrometric or micrometric particles, but not the actual particle size. For that, information on particle size distribution (for instance) must be retrieved.

**R#04:** The reviewer point of view is correct. Therefore, we accepted and expanded the description of Angstrom exponent as suggested.

**Q#05:** Page 9, line21: please include also the reference Moran-Zuloaga et al. (2018) Moran-Zuloaga, D., Ditas, F., Walter, D., Saturno, J., Brito, J., Carbone, S., Chi, X., Hrabe de Angelis, I., Baars, H., Godoi, R. H. M., Heese, B., Holanda, B. A., Lavri ˇ c, J. ˇ V., Martin, S. T., Ming, J., Pöhlker, M. L., Ruckteschler, N., Su, H., Wang, Y., Wang, Q., Wang, Z., Weber, B., Wolff, S., Artaxo, P., Pöschl, U., Andreae, M. O., and Pöhlker, C.: Long-term study on coarse mode aerosols in the Amazon rain forest with the frequent intrusion of Saharan dust plumes, Atmos. Chem. Phys., 18, 10055-10088, https://doi.org/10.5194/acp-18-10055-2018, 2018.

**R#05:** Suggestion accepted.

**Q#06:** Page 9, line 24: replace "aerosol optical depth" by "AOD" because you have already introduced the acronym. Check along the manuscript.

R#06: Suggestion accepted.

**Q#07:** Page 10, lines 5-6: Cloud screening is done based on the large standard deviations from the mean observed in some cases. However, it is necessary to quantify what you refer with "large standard deviations".

**R#07:** The text was rewritten including the description of the threshold used to apply cloud screening:

"*MFRSR AOD$_\lambda$ at 1-minute rate was averaged within a 5-minute interval centered on AERONET retrieval. Large standard deviations from the mean AOD, i.e. higher than 0.08 (considering 3x AERONET AOD products nominal uncertainty, which is 0.02), were interpreted as cloud contamination in MFRSR, therefore excluded from the analysis.*"

**Q#08:** Page 10, lines 8-9: replace the word "true" by "reference". Page 12, Table 1 and Table 2: Following equation 1 the slope (after changing its sign) provides information on the daily average of AOD. All the values shown in Table 1 and Table 2, both referred to 500 nm, are in the range 0.20-0.30, what from my point of view are not low enough to be considered as clear atmosphere or values similar to those obtained at a mountain top (above the atmospheric boundary layer). In this sense, more discussion is needed.

**R#08:**
"True" replaced by "reference"

The slope represents the daily average of total atmospheric optical depth (including molecular OD + gaseous absorption OD + aerosols OD). Average molecular OD in central Amazonia at 500 nm is 0.14, and Ozone OD 0.01. So, the daily average AOD would be in the range of 0.05 - 0.15, which is the range typically observed in Amazonia background atmosphere (Schafer et al., 2008).

This analysis was included in the text to clarify.

**Q#09:** Page 14, line 2 (but also this is an overall comment): Only years 2012 and 2015 are analyzed in this paper? What about 2013 and 2014? Is there any reason for this lack of information?

**R#09:** Please, see the reply for the question R#03.

**Q#10:** Page 15, line 17: You attributed the few suspicious points to the presence of optically thin cirrus. This can be easily checked from lidar measurements. Do you have simultaneous lidar measurements to corroborate this?

Indeed, there was a Lidar operating at T0e site. Using the data from this Lidar, Gouveia et al. (2017) showed that cirrus clouds classified as subvisible (Optical Depth < 0.03) frequency in Central Amazonia can be as high as ~ 42%, while thin cirrus (0.03< Optical depth < 0.3) ~38%. The presence of this subvisible and thin cirrus clouds very likely contaminate some AOD retrievals from MFRSR, Cimel and satellite sensors.

Therefore, we agree that Lidar analysis would contribute to the discussion of the present manuscript, however, as it is in course a study focusing on the comparison of AOD in Central Amazonia from multiple platform (ex. Cimel, MFRSR, MODIS), in which the Lidar information will be used, we evaluate that, while a Lidar data analysis would be interesting in the context of present manuscript, its absence would not jeopardize the major results of this manuscript. So, in the mentioned ongoing study (focused on AOD variability and not on sun photometer on site calibration) we will explore the role of seasonal and diurnal cycle of subvisible cirrus (as pointed out by Gouveia et al, 2017).

**Q#11:** Page 16, line 5: AERONET is not an instrument, replace by Sun-photometer or the AERONET Sun-photometer. The same in page 7, line 2.

**R#11**: Accepted

**Q#12:** Page 17, line 3: what about the results for 2013 and 2014? Page 17, figure 4: From this figure it seems that there is a overestimation of AOD from MFRSR respect to Cimel values, and underestimation of Angström Exponent values. Due to the different temporal resolutions I consider more convenient to present the temporal series of daily values or monthly values to check the overall coherence of both datasets. If monthly values are shown, the whole dataset (2012-2015) can be presented.

**R#12:** Regarding the results for 2013 and 2014, please, see **R#03**

To provide a consistent temporal series to compared AOD from both MFRSR and Cimel we replicate the Figure 7 (pag 17) including only coincident AOD and Angstrom Exponent retrievals (Please, see figure below). In this scenario, it is not clear that there is a systematic overestimation of AOD from MFRSR respect to Cimel, I would say that the MFRSR results are consistent with Cimel. Therefore, the apparent overestimation present in the Figure 4 is mainly related to MFRSR retrievals in scenarios when Cimel does not provide retrieval. In that case it is difficult to be conclusive about an eventual overestimation in MFRSR retrieval. It is recognized that AERONET AOD products are biased towards clear-sky conditions (Levy et al. 2010). Since cloudy conditions are likely to be related to distinct air mass conditions

(humidity and particles composition) and, therefore, different aerosol conditions, MFRSR would be able to capture high frequency of aerosol scenario not captured by Cimel. MFRSR higher frequency retrieval, mainly during the wet season (this can be seen in Figure 4 itself), improves the number of retrievals in scattered and broken cloud conditions.

In other hand, cloudy conditions may influence MFRSR AOD retrieval accuracy via the diffuse/direct radiation partition and, consequently, on the direct-normal irradiance used to obtain AOD. These are interesting aspects; however, their discussion is out of the context of the present manuscript, but we plan to address in the mentioned manuscript being prepared focusing on AOD variability in central Amazonia from multiple-platform(ground based and space based).

In the present manuscript, to clarify the apparent overestimation of MFRSR AOD respect to Cimel and highlight that it is not present when coincident retrievals of both devices are compared, Figure 4 was updated including an additional plot comparing only coincident retrievals (see figure below).

[Figure]

**Q#11:** Page 18, figure 5: All wavelengths should be shown here. Also, explain the meaning of asterisk on the y-axis unit (interpolated values, I guess), and the meaning of red dotted line (1:1 line).

**R#11:** All MFRSR channels able to provide AOD retrieval are shown (415 nm, 500 nm, 610 nm and 670 nm), only the channel 870 nm is not presented, which was justified previously due to the difficulty to obtain a consistent calibration for this channel.

The asterisk indicates that the AOD at that particular channel (Cimel) was estimated using Angström exponent, in that case to describe Cimel AOD. Red dashed line represents 1:1 line.

All these descriptions were included in the Figure 5 and 6 legends.

**Q#12:** Page 18, figure 6: The information from Figure 6 is summarized in Table 6. It would be nice if you replace figure 6 about AOD by the scatter plot of Angström exponents.

**R#12:** We think that the referee is meaning Table 4.

Since all information presented in Table 4 is already included in Figures 5 and 6 legends, instead of replacing 2015 AOD scatterplot by Angström exponent scatterplot in Figure 6, we evaluated to replace the Table 4 by a new Figure (see below) showing the scatter plot of Angström exponents considering the wavelengths 415 nm and 670 nm.

415 nm* indicates that the AOD used to obtain AE (415 nm*/670 nm) for Cimel was estimated using Angström exponent. (a) – year 2012; (b) year 2015

[Figure]

**Q#13**: Page 19, Table 4: All wavelengths and years must be shown here. Other table 7 might contain the corresponding information for Angström exponent

**R#13**: As explained in **R#12**, Table 4 is duplicating results already describe in Figure 5 and 6, so we replace it to a scatter plot of Angström Exponents (AE), therefore, we added information on AE without excluding any AOD information.

**Q#14:Technical corrections:**

Page 2, line 26: in ". . .the forest it is. . ." remove "it".

Page 7, line 7: replace ". . .using leas-square. . ." by ". . .using least-square. . .".

Page 15, line6: keep the same format along the paper (don't use I_(DN,λ) ).

Page 15, line 22: replace "one minutes frequency" by "1-min frequency".

Page 16, line 14: replace "sows and enhancement" by "shows an enhancement".

Page 17, caption figure 4: replace "Depth" by "depth".

Page 17, line 11: replace "For the 2015 years trends" by "The year 2015 trends".

Page 19, Table 4, caption: replace "aerosol optical depth" by "AOD" and "optical depth"by "AOD".

Page 19, line 7: replace "Do Central Amazonian pristine atmosphere provides" by "Does Central Amazonian pristine atmosphere provide".

Page 19, line 9: "Amazônia" Please, in English not in Portuguese.
Page 19, line 18: replace "varied" by "varies".

**R#14:** All recommended technical corrections listed below were accepted.

**References**

Schafer, J. S., T. F. Eck, B. N. Holben, P. Artaxo, and A. F. Duarte (2008), Characterization of the optical properties ofatmospheric aerosols in Amazônia from long-term AERONET monitoring (1993–1995 and 1999 – 2006), J. Geophys. Res., 113,D04204, doi:10.1029/2007JD009319

Kuhn, U., Ganzeveld, L., Thielmann, A., Dindorf, T., Schebeske, G., Welling, M., Sciare, J., Roberts, G., Meixner, F. X., Kesselmeier, J., Lelieveld, J., Kolle, O., Ciccioli, P., Lloyd, J.,Trentmann, J., Artaxo, P., and Andreae, M. O.: Impact of ManausCity on the Amazon Green Ocean atmosphere: ozone production, precursor sensitivity and aerosol load, Atmos. Chem. Phys., 10, 9251–9282, doi:10.5194/acp-10-9251-2010, 2010

Martin, S. T., Artaxo, P., Machado, L. A. T., Manzi, A. O., Souza, R. A. F., Schumacher, C., Wang, J., Andreae, M. O., Barbosa, H. M. J., Fan, J., Fisch, G., Goldstein, A. H., Guenther, A., Jimenez, J. L., Pöschl, U., Silva Dias, M. A., Smith, J. N., and Wendisch, M.: Introduction: Observations and Modeling of the Green Ocean Amazon (GoAmazon2014/5), Atmos. Chem. Phys., 16, 4785-4797, https://doi.org/10.5194/acp-16-4785-2016, 2016.

Suzane S. de Sá, Brett B. Palm, Pedro Campuzano-Jost, Douglas A. Day, Weiwei Hu, Gabriel Isaacman-VanWertz, Lindsay D. Yee, Joel Brito, Samara Carbone, Igor O. Ribeiro, Glauber G. Cirino, Yingjun Liu, Ryan Thalman, Arthur Sedlacek, Aaron Funk, Courtney Schumacher, John E. Shilling, Johannes Schneider, Paulo Artaxo, Allen H. Goldstein, Rodrigo A. F. Souza, Jian Wang, Karena A. McKinney, Henrique Barbosa, M. Lizabeth Alexander, Jose L. Jimenez, and Scot T. Martin
Atmos. Chem. Phys., 18, 12185-12206, https://doi.org/10.5194/acp-18-12185-2018, 2018

Gouveia, D. A., Barja, B., Barbosa, H. M. J., Seifert, P., Baars, H., Pauliquevis, T., and Artaxo, P.: Optical and geometrical properties of cirrus clouds in Amazonia derived from 1 year of ground-based lidar measurements, Atmos. Chem. Phys., 17, 3619-3636, https://doi.org/10.5194/acp-17-3619-2017, 2017.

John A. Augustine, Christopher R. Cornwall, Gary B. Hodges, Charles N. Long, Carlos I. Medina and John J. DeLuisi, An Automated Method of MFRSR Calibration for Aerosol Optical Depth Analysis with Application to an Asian Dust Outbreak over the United States, Journal of Applied Meteorology, 10.1175/1520-0450(2003)042<0266:AAMOMC>2.0.CO;2, 42, 2, (266-278), (2003).

MichalskyJ. J.J. A.SchlemmerW. E.BerkheiserJ. L.BerndtL. C.Harrison2001Multiyear measurements of aerosol optical depth in the Atmospheric Radiation Measurement and Quantitative Links programsJ. Geophys. Res.106D111209912107

Schmid, B, and C Wehrli. 1995. "Comparison of sun photometer calibration by Langley Technique and Standard Lamp." Applied Optics 34(21):4500-4512.

---

## Author Response (AR2)

**Authors considerations regarding the use of the information given in the responses to the Referees to improve the manuscript**

*Associate Editor Decision: Publish subject to minor revisions (review by editor) (22 Oct 2018) by Piet Stammes Comments to the Author:*

*"Dear Authors,*
*Thanks for your response to the referees. However, please use the information given in the responses to the Referees to improve the manuscript. This holds specifically for the responses to Referee #1: R#01, R#02 and R#03. For the response to Referee #2, this holds for R#03, R09 and R10. Currently it is not clear if anything is done with these useful comments from the referees."*

**We would like to thank the Associate Editor for his attentive on the points highlighted.**

**Response to Referee #01**

**R#01 (to Referee #1) –** We added the extraterrestrial response calibrations estimation based on median (**included in the Table 3, pag. 16**) and provided a discussion in the manuscript comparing the median results with the mean based values (**Pag. 14 from Line 7 to Pag. 15, line 7**).

**R#02 (to Referee #1) –** A explanation regarding the reasons to select the years 2012 and 2015 was included in the manuscript (**Pag. 8, Line 7 – 11**)

**R#03 (to Referee #1) –** Despite the restriction (majority of the individual Langley plot limited to the dry season) to provide an evaluation whether there is or not a seasonal dependence in the estimated extraterrestrial response, we added a discussion in the manuscript highlighting this aspect and, based on the referee hypothesis, observed that the stable temperature throughout the seasons in Central Amazonia is likely to diminish the seasonal dependence of the calibration constant.(**Pag. 15, Line 16 – 21**)

**Response to Referee #02**

**R#03 (to Referee #2) –** We believe that the addition in the manuscript related to the response **R#02(to Referre#1)** attend the editor consideration regarding **R#03 (to Referee#2).** (**Pag. 8, Line 7 – 11**)

Additionally, aiming to attend the Referee#2 query about whether our numbers of individual Langley plot are robust, we included short revision regarding the number o individual Langley plot typically in our support. (**Pag. 11, Line 20 to Pag. 12– 5**)

**R#09 (to Referee #2) –** Question#09 from Referee#2 resemble part of his own Question#03 and also Question#02 of Referee#1, both already considered above in **R#02 (to Referee #1)** and **R#03 (to Referee #2).**

**R#10 (to Referee #2) –** The cirrus potential to contaminate MFRSR and AERONET sunphotometer retrieval in Central Amazonia is better contextualized in the manuscript (**Pag. 17, Line 13 – 17**)

[revised manuscript text omitted]
 | N | $$ (mean) | $\sigma_{}$ (Std) | $$ (median) | N | $$ (mean) | $\sigma_{}$ (Std) | $$ (median) |
|---|---|---|---|---|---|---|---|---|
| | | **Year 2012** | | | | **Year 2015** | | |
| 415 nm | 21 | 1.586 | 0.015 (1.0 %) | 1.586 | 22 | 1.579 | 0.017 (1.0%) | 1.582 |
| 500 nm | 17 | 1.839 | 0.015 (0.8 %) | 1.829 | 21 | 1.870 | 0.015 (0.8%) | 1.890 |
| 613 nm | 14 | 1.545 | 0.010 (0.7%) | 1.537 | 17 | 1.572 | 0.011 (0.7%) | 1.592 |
| 670 nm | 15 | 1.416 | 0.010 (0.7%) | 1.405 | 18 | 1.433 | 0.008 (0.6%) | 1.443 |
| 870 nm | 15 | 0,842 | 0.008 (0.9%) | 0.846 | 20 | 0.802 | 0.003 (0.4%) | 0.804 |

**Comentado [NR1]:** In order to include Extraterrestrial calibration based on median we excluded the column related to the difference between calibrations based on mean for the years 2012 and 2015 since this can be easily inferred.

[revised manuscript text omitted]